# Investigation of Inflammation in Lewy Body Dementia: A Systematic Scoping Review

**DOI:** 10.3390/ijms241512116

**Published:** 2023-07-28

**Authors:** Paula M. Loveland, Jenny J. Yu, Leonid Churilov, Nawaf Yassi, Rosie Watson

**Affiliations:** 1Population Health and Immunity Division, The Walter and Eliza Hall Institute of Medical Research, Parkville 3000, Australia; 2Department of Medicine, The Royal Melbourne Hospital, University of Melbourne, Parkville 3000, Australia; 3Department of Neurology, Melbourne Brain Centre, The Royal Melbourne Hospital, University of Melbourne, Parkville 3000, Australia; 4Melbourne Medical School, University of Melbourne, Parkville 3000, Australia

**Keywords:** Lewy body dementia, Parkinson’s disease, α-synuclein, inflammation, biomarkers, neurodegeneration

## Abstract

Inflammatory mechanisms are increasingly recognized as important contributors to the pathogenesis of neurodegenerative diseases, including Lewy body dementia (LBD). Our objectives were to, firstly, review inflammation investigation methods in LBD (dementia with Lewy bodies and Parkinson’s disease dementia) and, secondly, identify alterations in inflammatory signals in LBD compared to people without neurodegenerative disease and other neurodegenerative diseases. A systematic scoping review was performed by searching major electronic databases (MEDLINE, Embase, Web of Science, and PSYCHInfo) to identify relevant human studies. Of the 2509 results screened, 80 studies were included. Thirty-six studies analyzed postmortem brain tissue, and 44 investigated living subjects with cerebrospinal fluid, blood, and/or brain imaging assessments. Largely cross-sectional data were available, although two longitudinal clinical studies investigated prodromal Lewy body disease. Investigations were focused on inflammatory immune cell activity (microglia, astrocytes, and lymphocytes) and inflammatory molecules (cytokines, etc.). Results of the included studies identified innate and adaptive immune system contributions to inflammation associated with Lewy body pathology and clinical disease features. Different signals in early and late-stage disease, with possible late immune senescence and dystrophic glial cell populations, were identified. The strength of these associations is limited by the varying methodologies, small study sizes, and cross-sectional nature of the data. Longitudinal studies investigating associations with clinical and other biomarker outcomes are needed to improve understanding of inflammatory activity over the course of LBD. This could identify markers of disease activity and support therapeutic development.

## 1. Introduction

### 1.1. Lewy Body Dementia

Lewy body dementia (LBD), the second-commonest neurodegenerative dementia after Alzheimer’s disease (AD), is characterized neuropathologically by abnormal α-synuclein aggregates (Lewy bodies and Lewy neurites) in the brain and clinically by progressive cognitive and motor impairment that leads to functional disability. Accurate diagnosis remains difficult, especially in early disease stages, due to the highly variable clinical course and lack of specific diagnostic tests [1].

LBD encompasses dementia with Lewy bodies (DLB) and Parkinson’s disease dementia (PDD), two syndromes with significant clinical and neuropathological overlap. They are distinguished by the timing of dementia onset related to motor Parkinsonism, which occurs within 12 months for DLB or greater than 12 months in PDD [2,3]. While not all people with Parkinson’s disease (PD) develop dementia, cognitive impairment is increased compared to the general population, and dementia risk increases over time, with a prevalence of dementia of up to 80% at twenty years from the initial PD diagnosis [4,5,6]. For clarity, in this review, ‘PD without dementia’ is referred to as ‘PD’.

The prodromal, or ‘pre-dementia’, stage of LBD is now an area of increased research focus towards understanding early disease mechanisms. Mild Cognitive Impairment (MCI) describes cognitive decline before significant loss of ability to perform usual social and occupational functions occurs, which is a defining feature of dementia. Specific research criteria for prodromal DLB have recently been proposed, including an operationalized definition of MCI associated with Lewy body pathology (MCI-LB) [7].

Despite advances in understanding of hallmark neuropathology and clinical disease stages and important learnings from related conditions, including PD and AD, key challenges remain. These include understanding the link between α-synuclein and pathogenic mechanisms of disease initiation and progression to dementia, establishing disease-specific biomarkers that relate to clinical outcomes, and identifying targets for still-elusive disease-modifying therapy. The potentially central role of inflammatory pathways in dementia pathogenesis, and therefore the measurement of inflammatory molecules as markers of disease activity, is one avenue of exploration towards addressing these challenges [8]. Mounting evidence for aging-related disruption of homeostatic mechanisms, leading to chronic inflammation that perpetuates this disruption, supports this line of inquiry in aging-related diseases such as dementia [9].

### 1.2. Inflammation and Lewy Body Disease

Evidence from in vitro or animal disease models, direct examination of human postmortem brain parenchyma, and in vivo biofluid or imaging studies suggests an important relationship between α-synuclein, inflammation, and LBD pathogenesis [10,11]. Advances in the tools available to interrogate these mechanisms in human disease have recently opened new avenues of investigation.

Components of the central nervous system (CNS) important in neurodegenerative disease inflammatory signals include resident glial cells—microglia and astrocytes (astroglia) (Table 1). Microglia are immune competent cells that are central to innate inflammatory responses. Resting in a ‘ramified’ state, when activated, they change to an amoeboid morphology associated with phagocytosis and the release of inflammatory signaling molecules, including inflammatory cytokines [12]. Therefore, activated microglia are often considered to represent inflammatory activity. While no specific fluid-based biomarkers have yet been identified, non-invasive assessment of microglial function in vivo has become possible with positron emission tomography (PET) imaging. Ligands for the 18-kDa Translocator Protein Receptor (TSPO, aka peripheral benzodiazepine receptor), expressed on the outer mitochondrial membrane of activated microglia, have been used to detect neuroinflammation in a range of neurodegenerative diseases over recent years [13]. In clinical studies of α-synucleinopathy, TSPO PET studies detect increased microglial activation in disease-relevant brain regions in early PD [14].

Astrocytes also have a central role in brain homeostasis, including contributions to inflammatory responses, and have been implicated in several neurodegenerative diseases [15]. The importance of astrocytes in LBD pathogenesis is suggested by their role in α-synuclein degradation at the synapse; therefore, decreased astrocytic activity early in the disease course may be a pathogenic mechanism of α-synuclein accumulation [16]. In brain tissue, astrocytes are identified using the molecular markers glial fibrillary acidic protein (GFAP), an astrocytic intermediate filament considered to be a marker of astrocytic activation or more general central nervous system pathology, and YKL-40 (i.e., Chitinase-3-like protein 1 (CHI3L1)), which is associated with astrocyte activation and synaptic degeneration [17]. GFAP and YKL-40 are also detectable in cerebrospinal fluid (CSF) and blood.

Adaptive immune cell populations, such as T and B lymphocytes, are often closely linked with innate immune system activity, particularly in chronic inflammatory states where they can be involved in perpetuating inflammatory responses. Lymphocytes respond to glial cell release of inflammatory molecules and antigen presentation functions, which trigger activation and trafficking from the periphery into the CNS [18]. T lymphocytes in particular have been implicated in the pathogenesis of α-synucleinopathies, with increased numbers observed in PD brains at postmortem [19], and peripheral T lymphocytes from people with PD are reactive against α-synuclein peptides [20,21]. Peripheral blood mononuclear cells from people with PD have also implicated abnormalities in the inflammation-associated transcription factor nuclear factor kappa B (NF-kB), in particular, loss of NK-kB/c-Rel anti-inflammatory and neuroprotective activity [22].

Cytokines and chemokines, inflammatory signaling molecules, can also be measured across a range of biological tissues. Pro-inflammatory molecules include interleukin (IL)-1β, IL-2, IL-4, IL-6, IL-8, tumor necrosis factor (TNF)-α, interferon (IFN)-γ, and monocyte chemoattractant protein-1 (MCP-1). IL-10 and TGF-β are associated with a more regulatory or anti-inflammatory environment. Improved understanding of these signals in neurodegenerative disease has been facilitated by advances in biomarker assay technology, such as electroluminescence-based multiplex assays and the single molecule array (Simoa), which allow accurate measurement of molecules in ultra-low concentrations in biofluid (CSF and blood) [23]. In PD, a meta-analysis found higher levels of blood inflammatory cytokines (TNF, IL-6, IL-1β, IL-2, and IL-10) and the clinical acute phase reactant C-reactive protein (CRP) [24] compared to controls. Pro-inflammatory peripheral cytokine profiles have also been associated with more rapid motor and cognitive symptom progression in early PD [25].

Other signals for altered inflammatory mechanisms in neurodegenerative disease and α-synucleinopathy come from studies of genes and gene expression, including transcriptomics. A mutation in TREM2 (which encodes triggering receptors expressed on myeloid cell 2 (TREM2), expressed by brain microglia, and involved in determining inflammatory phenotype) is a risk factor for PD as well as AD and frontotemporal dementia [26]. In a prospective study of community-dwelling older adults without dementia, higher levels of serum sTREM2 were associated with an increased risk of future development of all-cause dementia (which included DLB cases), AD, and vascular dementia [27]. A systematic review and meta-analysis of LBD gene expression studies identified several differentially expressed genes (DEGs) in LBD brain tissue and biofluid. Functional analysis identified DEG enrichment in neuroinflammation signaling pathway genes, including genes related to myeloid cell movement [28]. 

Considering the substantial interest in recent years regarding inflammation in LBD and the new in vivo opportunities brought by advances in research tools, we aimed to synthesize the current literature in this scoping review. A systematic scoping review methodology was selected to provide a comprehensive synthesis of this heterogeneous research field that is characterized by varying approaches and definitions in order to identify key themes and areas for future research [29].

## 2. Objectives

Primary objective:Review the methods used to investigate inflammation in LBD and how these are applied to conceptualize inflammation.

Secondary objective:Identify alterations in inflammation signals in LBD compared to people without neurodegenerative disease and compared to related neurodegenerative diseases, in particular AD and PD.

## 3. Methods

This review was conducted according to the Preferred Reporting Items for Systematic Reviews and Meta Analyses extension for Scoping Reviews (PRISMA-ScR) guidelines (Appendix A) [30], using a pre-specified protocol (not published).

A systematic search strategy was applied to identify relevant publications.

Inclusion criteria:Original peer-reviewed research study.People with clinically diagnosed dementia with Lewy bodies (DLB), Parkinson’s disease dementia (PDD), or Lewy body dementia (LBD) without specification of DLB or PDD. For postmortem studies, the use of international neuropathological criteria for diagnosis was sufficient.Inflammation is assessed either postmortem or in vivo.

Exclusion criteria:Studies where LBD participants have a comorbid inflammatory condition that might impact the assessment of inflammation, e.g., encephalitis, multiple sclerosis, inflammatory arthritis, or sepsis.People with a Lewy body disease (e.g., Parkinson’s disease) without dementia, including people with mild cognitive impairment (MCI).In interventional studies, assessments of inflammation are taken while receiving or after the intervention.Non-English language.

A systematic search was performed using the electronic bibliographic databases MEDLINE, EMBASE, PsycINFO, and Web of Science. No restrictions were placed on the search. Reference lists of eligible studies and review articles were also screened. 

The following search terms were used:1.(Lewy.mp AND dementia) OR dementia with lewy bodies.mp OR ((Parkinson Disease/OR parkinson’s disease.mp) AND dementia.mp) OR Parkinson’s disease dementia.mp

AND
2.inflammat* OR neuroinflammat* OR Neurogenic inflammation/OR microglia.mp

An example search strategy is provided in Appendix A. 

The final search was conducted on 26 January 2023. Search results were imported into the Covidence online review management system, which was also used for charting data from included studies into standardized data collection forms. 

Two authors (PML and JJY) independently screened titles and abstracts to select studies for full-text review. The authors were initially blinded to each other’s decisions, and then any disagreements were discussed. If consensus was not reached, the study was included for full-text review. The full text review was performed independently by the same two authors, initially blind to each other’s decisions. If an agreement was not reached through discussion between the two authors, the studies were discussed with two senior authors (RW, NY).

When multiple analyses or publications relating to a single research study cohort were identified, these were, where possible, combined for extraction with the aim that participant numbers were not counted more than once. If validation cohorts were reported, these were only included for data extraction if baseline cohort demographics were provided and all inclusion criteria were met.

## 4. Results

### 4.1. Search Results

After duplicates were removed, 2509 search results were screened, and 306 studies underwent full text review (Figure 1). Eighty study cohorts met the inclusion criteria. Study characteristics, inflammation assessment methods, and key findings relating to inflammation in LBD are summarized in Appendix A—Summaries of included studies (Appendix A).

### 4.2. Characteristics of Included Studies

Inflammation and LBD were investigated using postmortem brain tissue (36 studies), fluid or imaging biomarkers measured in living subjects (44 studies), and analysis of genetic factors (3 studies) (Table 2 and Figure 2). Most studies investigated inflammation in one substrate, with blood being the most common substrate analyzed, and only five studies reported on neuroimaging-based methods (i.e., PET) (Figure 3b).

All included studies provided cross-sectional data for participants with LBD, although two studies reported longitudinal data relating to participants with prodromal LBD [31,32] and one interventional study for PDD reported inflammatory outcomes at two timepoints [33]. Study sizes were small, with a median of n = 14 (interquartile range (IQR) 9–27.5) LBD participants per study, although in vivo studies generally had larger LBD group sizes (median n = 21.5 participants, IQR 10–34.5) than postmortem studies (median n = 10 participants, IQR 6.5–16). Median DLB and PDD group sizes were similar, with a median of n = 14 (IQR 9–25) and n = 18 (IQR 9.75–27.5) participants, respectively. 

Included studies report cohorts from Europe (United Kingdom, 19 studies; rest of Europe, 34 studies), Asia (12 studies), North America (19 studies), South America (1 study), and Australia (1 study), of which six studies included participants from multiple regions (Figure 3a). Participants with DLB were present in 53 studies; a PDD group was present in 29 studies; and 11 reported an LBD group without reporting separate DLB and PDD results. Direct comparison between DLB and PDD cases was uncommon, with only 12 studies including both diagnostic groups. Most studies reported pathobiological insights or discrimination from other diagnostic groups, especially AD, PD, or age-matched controls. A non-dementia control group was present in all but eight studies.

### 4.3. Investigation of Inflammation in LBD—Insights on Disease Pathobiology from across the Disease Timeline

Signals of altered inflammatory mechanisms in LBD appeared to be variable across the disease timeline (Figure 4). In prodromal and early established LBD, biofluid and functional PET imaging data suggest increased inflammation, whereas in late-stage disease, postmortem neuropathology suggests altered inflammatory profiles and possible immune senescence. Key components of innate and adaptive immune responses identified include CNS-resident glia (microglia, astrocytes), circulating helper T lymphocytes, and cytokines.

#### 4.3.1. Microglia—Activation or Dysfunction?

The understanding of the role of microglia in LBD disease mechanisms has evolved with the application of new analytical techniques and the re-interrogation of previous conclusions. 

#### Postmortem Investigation of Microglia

Analysis of LBD brain tissue by McGeer et al. provided the earliest reports associating microglia with hallmark neuropathology [34]. The hypothesis of an inflammatory innate immune response directed by microglia in LBD was supported in the next two decades by several small postmortem studies that reported increased microglial HLA-DR (human MHC II) expression, association of microglia with degenerating Lewy body-containing microglia, microglia engulfing Lewy bodies in late-stage disease, and increased expression of pro-inflammatory cytokines (IL-6 and TNF-α) [35,36,37,38]. A few contemporaneous studies found a lack of these signals in LBD brains [38,39,40], with Rozemuller et al. notably assessing microglia morphology in their determination of activated microglia populations, which were associated with AD plaques but not LBD pathology. 

Potential stimuli for LBD-associated glial cell activation identified at postmortem include the complement pathway [37,38,41], inducible nitric oxide synthase (iNOS) [42], NF-kB (a transcription factor activated by inflammatory cytokines), and the leucine-rich repeat kinase 2 (LRRK2) nuclear translocation of activated T-cells cytoplasmic 2 (NFATc2) signaling cascade [38]. Lack of replication in multiple cohorts or conflicting negative studies for some of these proposed mechanisms, such as complement activity [40], however, limit the conclusions that can be drawn. 

Three decades after the publication of McGeer et al., several groups also used microglial morphological assessment to argue against the initial proposition that microglial activation is a hallmark of LBD neuropathology. Classification of activated microglia by visual confirmation of an ameboid state and increased staining of activation markers, such as CD68, found no increase in activated microglia in LBD brains and, interestingly, the presence of greater numbers of dystrophic microglia compared to age-matched controls [43,44,45,46,47]. A decreased phagocytic capacity of microglia has also been identified in the hippocampus of DLB and PD brains, along with an age-related decline in microglial function in all study groups, including controls [48].

#### PET Imaging of Microglia In Vivo

Understanding microglial contribution to LBD has recently been progressed by in vivo PET imaging studies using the [11C]PK11195 ligand against the TSPO microglial marker, albeit in only a few small studies (3 of PDD, 2 of DLB; LBD group sizes n = 5–19). Several studies show an association between LBD pathology and increased microglial activity compared to controls on TSPO PET imaging [49,50,51,52], and two offer evidence for increased microglial activity in earlier disease stages (one each for DLB and PDD) compared to participants with more advanced LBD. Changes in [11C]PK11195 binding were observed between early and late LBD as disease was assessed by clinical outcome markers and brain imaging assessment of white matter integrity [53,54]. 

#### 4.3.2. Astrocytes—Associated with LBD Pathology at Postmortem but Difficult to Assess in Life

##### Postmortem Investigation of Astrocytes

The markers GFAP and YKL-40 have been used to identify astrocytes in LBD brain tissue and test the hypothesis of a pathological contribution to disease processes, which likely include inflammatory mechanisms based on known functions, including interactions with activated microglia [55]. Increased GFAP+ astrocyte numbers have been reported in several studies [56,57,58] and associated with extracellular Lewy body pathology [37,38] and intra-neuronal Lewy bodies, including IL-1α- and TNF-α-positive astroglia [42]. Low et al. also provide a potential mechanism for prolonged astrocyte activation and contribution to inflammation through upregulation of FynT in DLB in response to tauopathy, a tyrosine kinase isoform that mediates prolonged inflammatory responses and astrocyte activation [58]. Association with AD co-pathology was suggested in one study where YKL-40-positive astrocytes in DLB were associated with AD-type neuropathology [59]. YKL-40 mRNA and protein expression in DLB is unclear, with one study finding no difference despite increased numbers of astrocytes and another finding increased YKL-40 and GFAP expression compared to controls despite minimal Lewy body or microglial pathology [59,60]. Adding to the complexity of interpreting these markers, one study presented YKL-40 staining as a marker of alternative microglial activation but did not associate it with astrocytes [46]. 

##### Biofluid Assessment of Astrocyte-Associated Inflammation In Vivo

In living subjects with LBD, evaluation of astrocytes is limited to biofluid due to a lack of suitable imaging ligands. YLK-40 measured in CSF and blood has been presented as a marker of inflammation, often associated with astrocyte activity, in several LBD cohorts [53,59,61,62,63,64,65,66,67]. Most studies did not find altered YKL-40 levels compared to controls. Interestingly, AD groups usually had significantly higher levels of CSF YKL-40 compared to DLB, and in one DLB cohort, CSF YKL-40 levels were positively related to AD-associated tau pathology [64], suggesting that astrocyte activity, or at least expression of this related molecule, may be more relevant to AD neuropathology than α-synucleinopathy.

Regarding GFAP, measurement in CSF or blood has been applied in a few recent LBD cohorts; however, it has largely been considered a marker of general neurodegeneration and not linked to inflammatory mechanisms or astrocytes specifically, and therefore was not identified in the systematic search for this review. One study included in this review measured GFAP in biofluid along with other inflammatory markers in CSF and found that GFAP levels did not correlate with YKL-40 across diagnostic groups, including DLB [61]. 

#### 4.3.3. Lymphocytes and the Adaptive Immune Response in LBD-Associated Inflammation

##### Postmortem Investigation of Lymphocytes

T lymphocyte involvement, and particularly the CD4+ helper T cell subset, has been identified by a few small studies of advanced LBD. In postmortem DLB and PDD brains, increased recruitment of CD3+ or CD3+/4+ T lymphocytes has been observed compared to controls across multiple brain regions, including in association with glial cells and α-synucleinopathy [46,68,69,70,71]. One study also reported the assessment of CD3/CD8+ cytotoxic T lymphocytes and CD20+ B lymphocytes, which were either scarce or absent in DLB brains [69].

##### In Vivo Biofluid Assessment of Lymphocytes

In living cohorts, altered adaptive immune system activity in LBD was suggested by two studies, which found reduced activation of CD4+ T and B lymphocyte subsets in blood compared to controls and AD groups, with one also reporting a non-significant trend towards increased CD8+ cytotoxic T lymphocytes in DLB compared to controls and AD [72,73]. The study by Amin et al. correlated these lymphocyte differences with increased peripherally measured pro-inflammatory cytokines (IL-1β and IL-6), but there was no difference in age-adjusted pro-inflammatory TNF-α or anti-inflammatory IL-10 levels compared to controls. 

#### 4.3.4. Inflammatory Molecules in LBD: Cytokines, Vascular Mediators, and Others

In this review, several studies measured cytokines or other inflammatory mediators in postmortem or in vivo samples. Details of each study and key outcomes are reported in Appendix A, with the most consistent patterns and replicated results summarized below.

#### Cytokines and Chemokines—Postmortem and In Vivo Studies

While results are variable, differences between LBD and controls point towards altered inflammatory profiles. Interestingly, discordance between DLB and PDD in postmortem and in vivo studies was also noted. 

Regarding pro-inflammatory cytokines in DLB, altered TNF-α and IL-6 levels were detected in several cohorts at postmortem [36,42,56] and in CSF from living subjects [74,75]. In contrast, in PDD, there was no difference in these cytokines detected in postmortem brain tissue [70] or CSF in vivo [76]. Expression of the inflammatory chemokine IL-8 was no different from controls in a postmortem PDD study [70], higher in CSF from PDD participants [76], and lower in CSF from DLB participants compared to controls [75]. One study associated inactivation of anti-inflammatory homeostatic mechanisms with increased levels of all measured cytokines (IL-1α, IL-13, and IL-10) and amyloid-β pathology load in DLB brains but not PDD [77].

In blood, several recent studies applied large cytokine panels to look for differences between groups of interest with variable results, including possible associations with clinical outcomes [31,53,72,78,79]. Cytokines implicated in LBD as different from controls in blood were similar to those identified at postmortem and in CSF, including TNF-α, IL-1β, IL-6, IL-8, MIP-3a, and IL-17a [53,72,80,81], including one study of RNA expression levels [82]. The correlation of selected cytokines with DLB clinical disease features was reported in a few studies, including higher TNF-α (worse neuropsychiatric symptoms), higher IL-6 (worse cognitive performance) [80], and higher IL-10 (worse motor Parkinsonism) [83]. Other DLB cohorts using more extensive multiplex assays, however, found no correlation between cytokine levels and clinical assessment of cognition, neuropsychiatric symptoms, motor Parkinsonism, or microglial activation measured with [11C]PK11195 PET [54,72]. 

One cohort assessed blood cytokine profiles over disease stages by comparing pre-dementia DLB (mild cognitive impairment with Lewy bodies, MCI-LB) and established DLB [83]. They found increased pro-inflammatory (IL- 1β, IL-2, and IL-4) and anti-inflammatory (IL-10) cytokine levels in the early DLB group that were reduced to levels of healthy controls in more advanced disease. TNF-α levels were, however, higher in the DLB group compared to MCI-LB, and there was no difference in IL-6. This cohort was expanded and followed in a longitudinal study of people with prodromal Lewy body and Alzheimer’s disease for up to three years [31]. Changes in several inflammatory cytokines (IFN-γ, IL-1β, IL-2, IL-4, IL-6, and IL-10) were detected over time in the prodromal DLB and AD groups. Worsening cognitive impairment was correlated with a decrease in the cytokines IFN-γ, IL-1β, IL-2, IL-4, and IL-10. It is important to note that only 35 of the 58 MCI-LB initially recruited completed a second assessment. Baseline cognitive scores in the group that was lost to follow-up were significantly worse, suggesting the possibility of a more severe disease course, although there was no difference in baseline cytokine levels. Therefore, a true difference between DLB and AD inflammatory profiles in established or advanced disease may not have been detected. 

#### Vascular Inflammatory Mechanisms in LBD

Vascular endothelial cells can play important roles in inflammatory responses through the expression of inflammatory mediators, including cytokines and chemokines, that recruit circulating innate and adaptive immune system elements to sites of tissue injury. 

One European DLB cohort investigated in vivo several CSF inflammatory markers implicated in vascular endothelial inflammation, including their relationship with other inflammatory molecules [84,85,86,87]. These included angiotensin-converting enzyme (ACE), soluble intercellular adhesion molecule-1 (sICAM-1), soluble vascular cell adhesion molecule-1 (sVCAM-1), and soluble platelet endothelial cell adhesion molecule-1 (sPECAM-1). While all have been implicated in mechanisms of inflammation and immune system activation, ICAM-1 is upregulated by inflammatory cytokines and shown to be abnormally expressed in CNS pathologies, and VCAM-1 is particularly involved in the recruitment of leukocytes into tissues. Increased sICAM-1 was detected in both the blood and CSF of DLB cases compared to AD and controls, and VCAM-1 levels were lower in the CSF of DLB cases compared to controls. 

Concordant with these findings, ICAM-1 expression was higher in the DLB group but not the PDD group compared to controls in a postmortem study, although ICAM-1 did not correlate with visualized Lewy body pathology [88], and VCAM-1 was downregulated in a transcriptomics postmortem LBD brain tissue compared to controls [89].

#### Clinical Acute Phase Inflammatory Markers—CRP and Procalcitonin

Elevated CRP and high-sensitivity CRP (hsCRP) levels in PDD, compared to PD and controls, have been demonstrated in blood and CSF [76,90,91,92]. In contrast, blood hsCRP levels in DLB participants were not different from controls [53] or LB-MCI [83] groups. Whether this represents a true difference between PDD and DLB disease inflammatory signals is difficult to interpret given the small studies and non-specific nature of this inflammatory marker.

Another acute-phase clinical marker, procalcitonin, was evaluated in only one included study [93]. In the small DLB group (n = 8), CSF procalcitonin levels were increased compared to controls, but there was no difference detected in blood procalcitonin.

#### Other Inflammatory Molecules Measured Postmortem and In Vivo

A range of other inflammation-associated molecules have been assessed in only one or two small studies investigating inflammation in LBD. 

At postmortem, altered soluble epoxide hydrolase [94] and cyclooxygenase-2 [95], but not sphingolipids [96], have been reported in LBD brains.

In vivo studies suggest that osteopontin (CSF and blood) [97,98], serpins (CSF and blood) [85], sphingolipids and lysophospholipids (blood) [99], and serum amyloid A (CSF) [76] are altered in LBD disease cohorts compared to controls and therefore may have neurobiological relevance to LBD, or at least neurodegeneration more broadly. Protein glycosylation profiles might predict conversion to DLB versus PD clinical disease in people with idiopathic REM sleep disorder [32]. Cystatin C may be relevant to the risk of dementia in PD [92,100] or be able to reflect treatment response in PDD [33]. Altered complement factors in PDD may be related to AD co-pathology [101]. Measurement of CSF oligoclonal bands [102,103] or the glial-associated molecules TREM2 and progranulin [64] did not reveal any differences in LBD groups, whereas another study did find elevated soluble TREM2 (sTREM2) levels (blood and CSF) in the DLB group compared to controls [67]. 

#### 4.3.5. Other Approaches to Inflammation Assessment in LBD

##### Serum Small Extracellular Vesicles

Analysis of RNA expression in serum small extracellular vesicles (SEVs), an emerging technique that allows in vivo analysis of CNS-derived RNA that crosses into peripheral circulation, also provides evidence for altered inflammatory cytokine activity in DLB compared to controls [104]. This study identified differentially expressed genes (DEG) verified by high-throughput quantitative polymerase chain reaction (qPCR), including downregulation of proinflammatory genes IL1B, CXCL8, and IKBKB. Functional pathway and network analyses found significant enrichment of molecular pathways involving inflammation, with downregulation of proinflammatory pathways including IL-6, IL-8, IFN, and T-cell receptor signaling in DLB, leading to the conclusion that immunosenescence may be an important contributor to DLB pathology. 

##### Transcriptomics

Transcriptome-based approaches for the analysis of postmortem LBD brain tissue also implicated the importance of altered inflammatory pathways and immune senescence at late stages. In DLB compared to controls, DEGs included downregulation of inflammation and immune system gene clusters, including antigen presentation and processing (including MHC II) and the innate immune response activating the cell surface receptor signaling pathway [105]. Similarly, down-regulation of inflammation-related genes for IL-1β, CXCL11 (a chemokine), and VCAM-1 was identified in LBD groups compared to controls [89]. Dysfunctional molecular network analyses suggested an important role for immune senescence, along with oxidative stress and mitochondrial dysfunction, in the development of LBD pathology. In a study of pulvinar tissue from DLB cases where Lewy body neuropathology burden was low, whole transcriptome RNA sequencing identified several enriched pathways compared to controls, from which the authors focused on genes relating to positive immune system processes and validated these findings with the detection of higher protein levels of astrocyte markers (GFAP and YKL-40) and the cellular stress-associated molecule heat shock protein 70 1A (HSPA1A) in DLB [60].

One study of mRNA expression from whole blood collected in vivo found upregulation in TNF-α signaling via the NFκB pathway and the Inflammatory response pathway in DLB (although not statistically significant) and AD on gene set enrichment analyses [106]. Differences in IFN-α and IFN-γ response pathways between DLB and AD were also detected.

##### Genetic Factors

Two studies evaluated genetic variants relevant to cytokines in LBD (Appendix A). One study of people with PD carrying heterozygous mutations in the GBA gene, a single gene mutation associated with sporadic PD, found increased IL-8 levels associated with worse cognition across the whole cohort (which included PD-MCI and PDD) [107]. Baseline demographics between PD, PD-MCI, and PDD, however, were not reported. Therefore, potential confounders and relevance to other cohorts could not be assessed. Polymorphisms in promoter regions of IL-10 and IL-18 genes, selected due to previously identified relevance in PD and cognitive impairment, were not associated with risk of PD compared to healthy controls or with cognitive impairment within a PD/PDD Han Chinese population [108].

Another study looked to validate rare coding variants ABI3_rs616338-T and PLCG2_rs72824905-G that had been identified as important for AD risk and implicated in alterations in neuroimmunity and neuroinflammation for disease pathogenesis [109]. While the only significant association across multiple neurodegenerative diseases was found for the AD group, a trend towards significance was noted for DLB. The impact of AD co-pathology on the DLB group result could not be assessed as only 67/306 DLB cases were neuropathologically verified.

## 5. Discussion

From early to advanced-stage disease, altered inflammatory signals have been detected in LBD. Previously considered a bystander consequence of neurodegeneration-driven toxicity, studies identified in this systematic scoping review examined inflammation as a potential driver of disease progression and marker of disease activity. Advances in investigation techniques have allowed interrogation of pathogenic inflammatory mechanisms proposed by in vitro and animal models of α-synucleinopathy with human studies [10,110]. 

### 5.1. Contribution of Inflammatory and Related Immune System Components to LBD Pathogenesis

Initiation of innate inflammatory mechanisms through α-synuclein-related glial cell activation and inflammatory cytokine production is one proposed mechanism of neurodegeneration in LBD. In this review, alterations in microglia numbers and pro-inflammatory activity were seen in several studies of postmortem brain tissue or in vivo analysis of biofluid and PET imaging. These findings are supported by pre-clinical studies that show increased α-synuclein translocation into neuronal vesicles in stress states, where it abnormally aggregates and can be released by exocytosis [111]. Soluble α-synuclein aggregates can trigger neurotoxicity via direct cell-autonomous mechanisms (mitochondria, vesicle recycling, and protein homeostasis abnormalities) and indirect non-cell-autonomous mechanisms involving transmission of α-synuclein between neurons and glia and extracellular α-synuclein [57,111]. Microglial activation via toll-like receptors (TLRs), in particular TLR2 and TLR4, results in the production of inflammatory cytokines including TNF-α, IL-1β, and IL-6 [112,113,114,115,116,117]. Enhanced activity of the multifunctional kinase enzyme LRRK2 in microglia is an important link between α-synuclein exposure, TLR2, and the release of proinflammatory cytokines (TNF-α and IL-6), through modulation of NFATc2 [57,118]. Reduced DNA-binding activity of the transcription factor NF-kB/c-Rel may also contribute to the perpetuation of inflammation in α-synucleinopathies [22]. 

The variability in microglial profiles over the disease course is an important recent development in the study of LBD pathobiology. In the human studies reviewed, a case can be made for microglial activation in early disease, which may be initially neuroprotective or enhance vulnerability to degeneration, followed by a chronic inflammatory state and subsequent microglial dystrophy and loss of function. PET imaging studies showing reduced microglial activity associated with more advanced disease are congruent with postmortem findings of dystrophic microglia in late-stage disease, which could result from chronic inflammation that has been present for many years at the time of death. While longitudinal human data is required to confirm this pattern of early microglial activity that wanes with disease progression, in a rat model of α-synucleinopathy immunoreactive microglia numbers peaked at 2 months from disease induction in association with Lewy body-like neuronal inclusions, 3 months prior to neurodegenerative changes (nigral dopamine neuron loss) [119]. Microglial populations derived from peripheral macrophages that infiltrate the CNS or reside on the border with the peripheral circulation have also recently been identified as potentially important mediators of the chronic inflammatory response to α-synuclein [120]. Pathogenic mutations in *LRRK2*, a PD-risk gene, have also been linked with increased α-synuclein-related CNS infiltration of pro-inflammatory peripheral monocytes in a mouse model [121].

Altered longitudinal inflammatory cytokine profiles identified in prodromal Lewy body disease may also be related to this microglia-related disease mechanism, although microglial-specific markers in blood or CSF have not been identified. An important consideration is that the signature of activated microglia is not unique to LBD. This is also observed in other neurodegenerative dementias at postmortem and on PET, including AD, FTLD, and PSP [122]. Therefore, while these findings of altered glial activity in LBD provide important pathobiological insights, LBD-specific mechanisms still remain to be elucidated, and these signals are less useful for clinical diagnostic settings where the most pressing issue is often to distinguish between neurodegenerative syndromes. 

Astrocytes, another glial cell population integral to CNS immune responses and widely investigated in neurodegenerative diseases, were also identified in this review as likely contributing to inflammatory signals in LBD. Questions remain, however, regarding whether astrocytes are key drivers of pathological inflammatory mechanisms of disease in LBD and also whether the currently utilized markers GFAP and YKL-40 accurately detect pro-inflammatory activation states of astrocytes when measured in CSF or blood [55]. GFAP levels track with neurodegenerative biomarkers in AD, predict conversion from MCI to AD, and increase longitudinally [17,123,124,125]. Therefore, GFAP may not distinguish CNS inflammation from neurodegeneration in dementia. In LBD biofluid studies, GFAP levels in blood and CSF are higher than controls, similar to AD, variably different from other neurodegenerative diseases, and may be associated with cognitive performance [126,127,128,129,130]. The correlation of GFAP with other in vivo inflammatory markers might help unravel this question. In contrast, YKL-40 measured in biofluid revealed a different signature, with levels in the LBD group being usually lower than those in AD groups and similar to controls. YKL-40 in LBD appeared to correlate with AD co-pathology, therefore raising the question of whether astrocytes are more active in AD compared to LBD or if there is some other amyloid or tau-mediated effect on its expression. It is feasible that astrocyte contribution to early inflammatory events in LBD is occurring; however, the currently applied techniques have not been able to detect this. Pre-clinical studies show that pro-inflammatory reactive microglia can activate astrocytes into a neurodegenerative-disease-promoting phenotype, which has also been visualized in PD and AD brains [131]. Alternatively, the role of this cell population in LBD pathogenesis may be minor compared to microglia and other factors.

T lymphocytes, particularly CD4+ helper T cells, may contribute to chronic inflammatory processes in LBD. In human studies and mouse models of α-synucleinopathy, their numbers are increased in parenchyma and perivascular brain regions, and they are located close to astroglial processes in areas of tissue with signs of glial activation and inflammation [69]. Pre-clinical models and human PD studies demonstrate an important role for α-synuclein-specific T cell responses in inflammation and neurodegeneration [20,132]. Investigation of the activation states of lymphocytes in peripheral blood is an emerging strategy in LBD, with a few recent studies using this approach to demonstrate differences compared to controls and AD cases. Despite this limited data, further investigation of lymphocyte contributions to LBD neuropathogenesis could be an important clue towards understanding the connection between innate and adaptive immune activity in LBD that perpetuates chronic inflammation. Considering the known functions of CD4+ T lymphocytes, glial presentation of Lewy body-related antigenic material in MHC II molecules and release of pro-inflammatory cytokines may recruit and activate peripherally circulating CD4+ T helper cells into the CNS [18,133]. CD4+ T helper cells play an important role in activating the immune response of other lymphocytes through cytokine secretion; therefore, the observed T cells may be contributing to the perpetuation of inflammatory activity and recruitment of other lymphocyte populations in LBD that are as yet undetected. 

### 5.2. AD Co-Pathology and Inflammation in LBD

The impact of AD-type co-pathology, hyperphosphorylated tau (p-tau), and amyloid-β may be important in the inflammatory signals detected in clinical LBD studies. The relationship between inflammation and hallmark proteinopathy is much more developed in dementia due to AD [134,135], where appreciated changes in the inflammatory signal over the disease course include early microglial activation and late-stage anti-inflammatory profiles [136,137,138]. While a high prevalence of AD co-pathology has been well described in DLB and, to a lesser extent, PDD [139], whether these mechanisms are relevant to LBD pathogenesis remains to be determined. In this review, the prevalence of AD co-pathology was only reported in a handful of studies but might impact levels of certain inflammatory molecules, such as YKL-40. Future studies assessing inflammation in LBD should also utilize markers of amyloid or tau co-pathology to better understand this dynamic.

### 5.3. Differences between the Lewy Body Dementias—DLB and PDD

While DLB and PDD are separate clinical diagnoses distinguished by the somewhat arbitrary assessment of a 12-month gap between motor Parkinsonism and dementia onset for PDD, significant overlap in hallmark neuropathology, genetics, imaging, and clinical features allows classification together under the umbrella of LBD. The debate continues as to whether these are truly distinct diseases or different spectrums of the same entity [140]. 

This review identifies some discordant signals in inflammatory profiles between DLB and PDD from different cohorts; however, varying methodologies between studies and a lack of direct comparison between DLB and PDD groups in all but 12 studies limit the conclusions that can be drawn. Future studies should directly compare DLB and PDD groups to clarify whether these conditions have truly different pathobiological inflammatory profiles and consider whether it is appropriate to combine PDD and DLB groups in future studies of LBD. Cohort studies of PD should also consider reporting PDD results separately, which will assist in understanding if there is an inflammatory signal specific to dementia development.

The hazards of divided research efforts in the investigation of LBD disease should also be considered. This review identified a significant body of work investigating inflammation in DLB or PDD, but most studies report on only one or the other diagnostic group. This is despite pre-clinical disease models, which form the basis of tested hypotheses, being applicable to both DLB and PDD. Given the challenges of performing adequately powered clinical studies in this often frail and multimorbid population of older adults, this is an important consideration. Future research efforts should focus on collaboration between DLB and PDD cohorts globally, using standardized research methods to enable comparison, to give the best chance of meaningful advances in understanding the inflammatory mechanisms of LBD disease pathobiology. 

### 5.4. Limitations of Identified Studies 

The discordance between clinical and neuropathological diagnoses in dementia, particularly LBD, is well documented [141]. A major limitation of current in vivo studies of LBD, which must be addressed as a priority to support the next stage of LBD discovery, is the lack of diagnostic confirmation with reliable markers of the hallmark α-synuclein proteinopathy [142]. This is in stark contrast to the field of AD research, where PET, CSF, and blood-based biomarkers for amyloid-β and p-tau support diagnostic confidence for clinicians and researchers [143,144]. While dopaminergic PET imaging ligands are available to support LBD diagnosis, they are not widely available globally and have limited sensitivity [2]. Previous attempts at ELISA-based measurements of α-synuclein in CSF and blood have failed to produce reliable results [145]. Keen interest now surrounds the measurement of aggregated α-synuclein using real-time quaking-induced conversion (RTQuIC) seeding assays, which have been successfully demonstrated for the brain, CSF, skin, submandibular gland, gut, and mucosal tissue [146,147,148]. In clinical LBD studies, CSF α-synuclein measurement by RTQuIC detects cortical Lewy body disease [149] and predicts the conversion of prodromal idiopathic RBD to clinical Lewy body syndromes [150]. Automation and reproducibility need to be established for more widespread application of RTQuIC-based α-synuclein assessment, and blood-based assays are even more preliminary [151]. However, this is a promising development towards neuropathological confirmation of clinical LBD in vivo.

Regarding study design, cross-sectional data and small sample sizes limit the conclusions that can be drawn from the studies identified in this review. Participant cohorts were also largely concentrated in Europe and North America; therefore, future research should include more geographically diverse populations that are followed longitudinally. Longitudinal studies including pre-dementia or ‘prodromal’ participants whose subsequent conversion to clinical dementia is not confirmed also must be interpreted with caution. Not all people with prodromal Lewy body disease go on to develop dementia, and therefore they do not all represent a component of the dementia disease spectrum. This is particularly important in the context of increasing research focus on the prodromal stage of neurodegenerative disease as a key target for disease-modifying therapy.

#### 5.4.1. Limitations of Postmortem Studies

Postmortem analysis of LBD brains provides invaluable insights into neuropathology at the tissue level; however, this method usually limits researchers to end-stage disease. Therefore, processes important in early disease development may be absent. While many brain banks now provide the clinical history of donors, the severity of dementia and clinical stage at death are often not available. Given the evidence from in vivo studies of a changing inflammatory signal across LBD disease stages, postmortem studies should endeavor to correlate histopathological data with clinical disease stage at death and compare signals in early and late-stage disease.

The anatomical region from which brain tissue was sampled must also be noted when evaluating postmortem studies, given the known regional distributions and variable progression rates of neurodegenerative pathological change in LBD. This was highly varied across the studies identified in this review (Appendix A). Future postmortem evaluations of potential inflammatory processes should sample multiple brain regions of significance. 

#### 5.4.2. Limitations of In Vivo Studies

The assessment of non-specific inflammatory markers in a disease cohort comprised of older adults poses several challenges. This demographic is more likely to have multiple comorbidities that may influence inflammatory signals, particularly those measured peripherally in the blood. The potential impact of comorbid neuropathology, such as cerebrovascular disease or AD-related pathology (amyloid-β and tau), must also be considered, the latter of which is known to be frequently present in DLB [152,153]. This could be somewhat mitigated by the assessment of these co-pathologies with imaging and biofluid markers, which have been well established for the evaluation of vascular and Alzheimer’s disease pathologies [143]. 

Biological age has also been shown to impact cytokine levels [24,76], microglial function [48], and inflammatory/immune system activity, so-called ‘inflamm-aging’ [65,75]. This concept was also demonstrated in the context of an animal model of Lewy body disease, where aged mice treated with the same induction methods of α-synucleinopathy showed greater microglial activation, T cell infiltration, and more extensive neuropathology compared to young mice undergoing the same treatment [154]. Future studies should report detailed demographic and comorbidity data for all participants and consider these in their analyses.

While the prospect of reliable biofluid-based markers is enticing given the potential for repeatability, transportability, and ease of collection, particularly associated with blood, additional potential confounders must be considered. Whether or not the molecule of interest crosses the blood-brain barrier and can be reliably measured in blood to reflect CNS pathology must be interrogated. In a few studies that investigated paired CSF and blood samples, no correlation was found between the measurements for several inflammatory molecules (IL12 p40, IL-12 p70, IFN-γ, TGF-β1, IL-10, and procalcitonin), albeit in early studies where assay sensitivity may have been lacking [93,155]. This is in contrast to studies testing neurodegeneration biomarkers such as NfL, GFAP, and p-tau, where a good correlation between CSF and blood levels has been demonstrated [129]. 

It is also important to note that in LBD, there was no difference found for the majority of inflammatory molecules assessed by studies in this review. This may be due to a difference only in a few key inflammatory mediators at various disease stages or the limitations of various assessment techniques and patient factors that may distort the detection of the true relationships. Several relatively recent studies using assays with improved sensitivities were still unable to report on planned biofluid cytokines due to undetectable or unreliable levels (TNF-α, IFN-γ, IL-1β, IL-6, IL-10, IL-17, etc.) [74,156], levels below the threshold of detection requiring analysis as a binary value (IL-1β in blood) [72], or high inter-assay variability (IL-13 and IL-12p70) [83].

Variations in methodologies used to analyze markers of inflammation may impact results. Blood levels of cytokines, and possibly other biomarkers of interest, are affected by diurnal variation [157]. Only a handful of identified studies, however, report standardizing collection time, with samples for inflammatory molecule analysis taken in the morning or fasted (likely a morning sample) [33,62,67,72,76,78,99,100,101,156]. As mentioned above, the sensitivity of the assays used may be too low to detect molecules of interest, particularly when comparing older studies to those using more advanced detection methods or comparing different assay platforms; therefore, conclusions that can be drawn about their absence are limited [158,159]. 

## 6. Conclusions

This systematic scoping review describes a range of approaches applied to the assessment of inflammation in LBD across several substrates, most commonly postmortem brain tissue or biofluid collected in vivo, with a few studies utilizing a PET imaging ligand for microglia. Identified studies reveal important insights into inflammatory mechanisms across the LBD disease course that show differences compared to control and other neurodegenerative disease groups. 

Changes over the course of the disease in LBD may be early pro-inflammatory events relating to innate immune system activation with contributions from microglia, astrocytes, and inflammatory mediators including cytokines and vascular mediators, some of which may be neuroprotective. Helper T lymphocytes may also become involved, linking innate and adaptive immune activation and contributing to chronic inflammation. In advanced stages of LBD, chronic inflammation and immune senescence involve dystrophic microglia and possibly reduced lymphocyte responses. Potential differences between PDD and DLB also require further investigation and replication.

Highlighted in this review is the challenge posed by the ubiquitous nature of key inflammatory and immune mechanisms and defining abnormal pathobiology compared to normal homeostatic pathways in neurodegenerative disease. As new molecules have been identified or their functions described in new ways, and the complex links with the immune system and other metabolic functions across human diseases are delineated, the interpretation of experimental results must be reviewed and potentially re-interpreted. This may yield novel approaches to disease detection, prognosis, and monitoring. Whether these inflammatory signals will yield a therapeutic target for disease modification with an acceptable adverse event risk profile, however, remains to be seen. 

Verification of these hypotheses will require significant additional data from well-characterized cohorts, with longitudinal measurements that correlate multiple inflammatory measurements and neuropathological verification at postmortem or in vivo. Application of ultra-sensitive assays for inflammatory molecules of interest may improve clarity. The impact of comorbid neuropathology, particularly of the AD type, must also be determined. Development of further imaging ligands to assess other CNS inflammatory signals beyond microglia is needed, as is the development of blood-based biomarkers that can be easily repeated over the course of the disease to track disease activity and, potentially, response to disease-modifying therapies in the future.

## Figures and Tables

**Figure 1 ijms-24-12116-f001:**
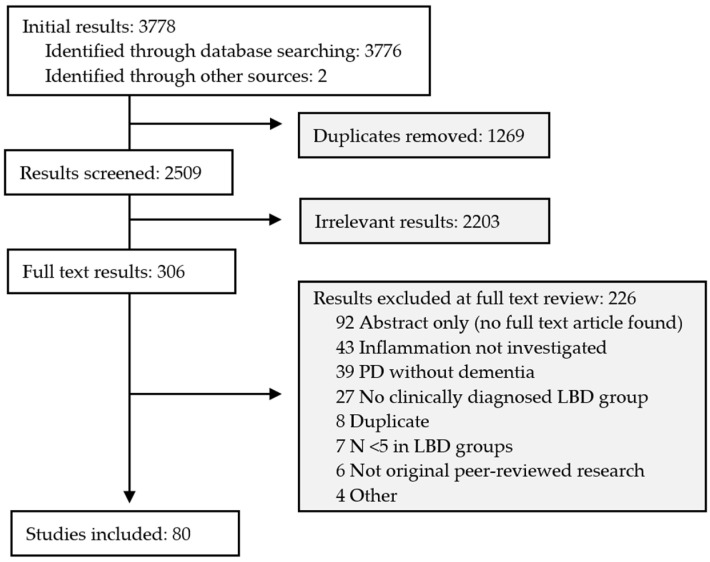
PRISMA-ScR flow diagram.

**Figure 2 ijms-24-12116-f002:**
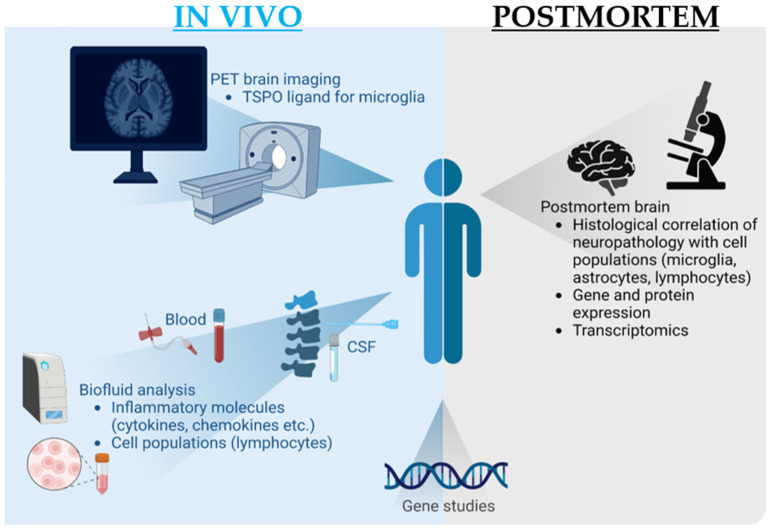
Methods of inflammation investigation in Lewy body dementia. A range of analytical techniques have been applied to the investigation of inflammation in Lewy body dementia (LBD). Postmortem brain tissue analysis includes visualization of neuropathology alongside immune cell populations with known inflammatory functions (microglia, astrocytes, and lymphocytes) and quantification of inflammatory mediators (e.g., cytokines) with RNA and protein expression studies. Several transcriptomics studies of LBD postmortem tissue also provide insights into potential inflammatory disease mechanisms. In living subjects with LBD, the inflammatory activity of microglia in the brain can be assessed with ligands against the 18-kDa Translocator Protein (TSPO) receptor. Biofluid-based analysis methods include measurement of inflammatory molecules and peripheral lymphocytes in both blood and cerebrospinal fluid (CSF). Interrogation of potentially relevant genetic factors in LBD-associated inflammation has been performed with tissue from postmortem and in vivo studies.

**Figure 3 ijms-24-12116-f003:**
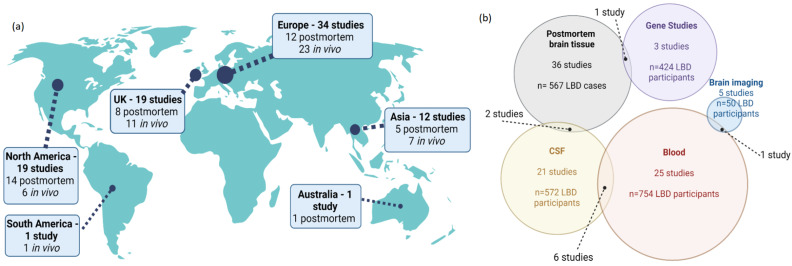
(**a**) Geographic distribution and (**b**) substrate type analyzed in the included studies. (**a**) Of the 80 included studies, 53 report cohorts from Europe. Within Europe, 19 studies included cohorts from the United Kingdom (UK); nine were from Spain; seven each from Germany and Sweden; four each from Norway and Italy; two from the Netherlands; and one each from Greece, Luxembourg, and Russia. In North America, 16 studies included populations from the United States of America, and three were Canadian. In Asia, six studies included Japanese cohorts; four were from China; and two were from Korea. One study included subjects from Brazil and one from Australia. NB: Six studies used cohorts from multiple countries, and two studies reported postmortem and in vivo samples; therefore, these are counted multiple times. (**b**) Inflammation in Lewy body dementia (LBD) was investigated by analysis of tissue from postmortem brains in 36 studies, of which two also reported CSF results from living subjects. Blood (serum or plasma) biomarker analysis was reported in 25 included studies comprising 754 participants with LBD, and 21 studies analyzed CSF from 379 LB participants. Six of these studies included both blood and CSF results. Brain imaging to assess inflammation was performed in five studies with 50 LBD participants, of which one also analyzed blood. Genetic studies were performed on DNA extracted from postmortem brain tissue (one study) and blood (three studies), although only one study also reported other markers of inflammation, which were from postmortem tissue.

**Figure 4 ijms-24-12116-f004:**
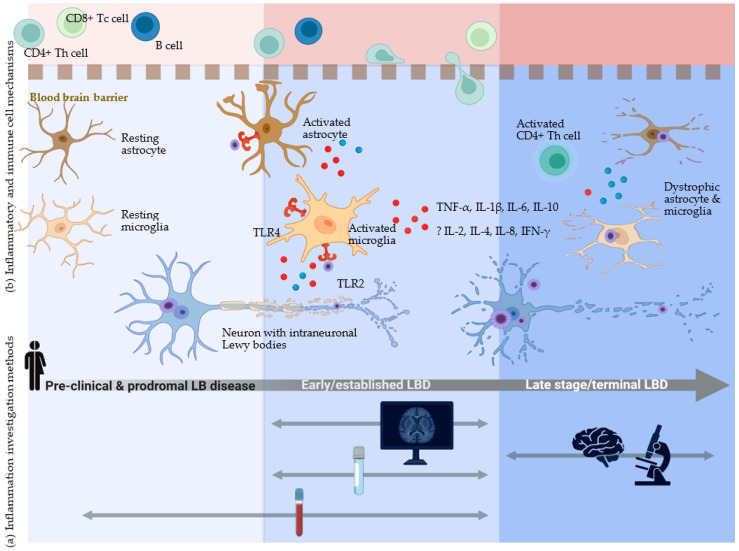
Across the disease timeline of Lewy body dementia: (**a**) methods of inflammation assessment and (**b**) proposed inflammatory mechanisms. (**a**) Tools applied to the investigation of inflammation in clinical Lewy body dementia (LBD) studies provide data from across the disease spectrum. Blood samples were collected from people with prodromal Lewy body disease before dementia was established. Blood, cerebrospinal fluid, and brain imaging data have been analyzed in early and established Lewy body dementia. Postmortem brain tissue samples are generally limited to people with end-stage LBD, although clinical disease severity at death is not always reported. These clinical research methods inform our understanding of LBD pathogenesis and the interpretation of data from pre-clinical models and related neurodegenerative diseases. (**b**) Inflammatory and immune cell mechanisms of inflammation in LBD are altered across disease stages. Prior to LBD development, resident brain glial cells (microglia and astrocytes) existed in resting states while performing homeostatic and surveillance functions. The accumulation of abnormal α-synuclein, which aggregates as intraneuronal Lewy bodies, likely triggers activation of innate inflammatory responses in early clinical LBD disease stages. One mechanism is via toll-like receptor (TLR) activation of microglia, which change from resting ramified to activated amoeboid morphology and express inflammatory cytokines including TNF-α, IL-1β, and IL-6. There is also evidence of altered levels of the anti-inflammatory cytokine IL-10 in LBD and possible differences in other inflammatory molecules such as IL-2, IL-4, IL-8, and IFN-γ. Astrocytes also become activated, produce inflammatory cytokines, and interact with the blood-brain barrier. These inflammatory signals from CNS glial cells may recruit circulating lymphocytes into brain tissue, particularly CD4+ helper T cells, and contribute to the perpetuation of chronic inflammation. In late-stage LBD, abnormal dystrophic microglia and astrocytes are observed, which may represent immune senescence. DLB: dementia with Lewy bodies; IFN: interferon; IL: interleukin; LB: Lewy body; LBD: Lewy body dementia; Tc cell: cytotoxic T lymphocyte; Th cell: helper T lymphocyte; TNF: tumor necrosis factor; TLR: toll-like receptor.

**Table 1 ijms-24-12116-t001:** Inflammatory and immune system components implicated in neurodegenerative disease-associated inflammation.

Glia	Tissue Markers
Microglia	▪ CD68 ▪ HLA-DR/MHC II ▪ Iba1 ▪ TREM2 ▪ TSPO
Astrocytes	▪ YKL-40, aka CHI3L1 ▪ GFAP ▪ S100B
**Lymphocytes**	**Tissue markers**
T cells	▪ CD3+ ○ CD4+: helper T cells (Th); sense antigens presented by MHC II molecules ○ CD8+: cytotoxic T cells; sense antigens presented by MHC I molecules
B cells	▪ CD19+ ▪ CD20+
**Signaling molecules**	**Molecules**
Cytokines and chemokines	Pro-inflammatory: ▪ TNF-α, IL-1β, IL-2, IL-4, IL-6, IL-8, IL-17A, IFN-γ, MCP-1 Anti-inflammatory/regulatory: ▪ IL-10, TGF-β

**Table 2 ijms-24-12116-t002:** Characteristics of included studies.

Investigation Method	Number of Studies	DLB, n	PDD, n	LBD NOS, n	Total LBD, n	AD, n	PD, n	Controls, n
Postmortem	36	334	151	82	567	242	183	587
Living subjects	44	608	493	130	1231	1641	1623	2614
Imaging	5	25	25	0	50	18	6	61
CSF	21	313	163	96	572	1292	801	1671
Blood	25	390	330	34	754	672	1045	1097
Gene studies	3	316	108	0	424	2770	955	3834
**Total ***	**80**	**1236**	**698**	**172**	**2106**	**4542**	**2661**	**6801**

* Some studies used multiple investigation methods, e.g., postmortem brain tissue and in vivo biomarkers or in vivo CSF and blood analysis; therefore, there is overlap in total study numbers and total participant counts. Where studies performed analyses on a smaller subset of their total study population, the total study population group size is reported here, and details of smaller subset analyses are provided in the Appendix A). Total LBD column reports combined a total of LBD NOS, DLB, and PDD. AD: Alzheimer’s dementia; CSF: cerebrospinal fluid; DLB: dementia with Lewy bodies; LBD: Lewy body dementia; LBD NOS: LBD not otherwise specified, i.e., participants are classified as LBD without specification of DLB or PDD diagnosis; PD: Parkinson’s disease (without dementia); PDD: Parkinson’s disease dementia.

## Data Availability

Not applicable.

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
