# Peer review of "Investigation of Inflammation in Lewy Body Dementia: A Systematic Scoping Review"

_ijms, 2023, doi:10.3390/ijms241512116_

Round 1

Reviewer 2 Report

This is a fantastic systematic scoping review that perfectly focuses on the LBD-related neuroinflammatory changes ranging from brain imaging studies, postmortem analysis, and in vivo biofluids detection. These clinical data are highly instructive for successful animal model development and in vitro drug screening platform establishment. This review is excellent because it tries its best to differentiate the inflammatory profile between the early vs late (advanced) stage of LBD and DLB vs PDD. We would like to bring up more questions for authors to consider adding into to make this fantastic review more self-explained and more comprehensive.

1: In review, the author mentioned several times the early vs advanced (or late) stage. We suggest the author illustrate more on the classification to help the reader understand if the same criteria are applied here. For example, MCI=early stage vs Dementia=advance stage (clinical diagnostic) or based on MoCA score.

2: In lines 83-85, the author mentioned the important role of astrocytes in early synapse synuclein degradation at the early stage of the disease. What makes astrocytes so unique on this other than microglia? Would the author add more explanation here?

3: In lines 344-359, the author suggested the current studies indicated that there is more CD4+ T cell infiltrated into the brain of DLB and PDD brains, but there are more (non-significant trend) activated CD8+ T cell and less CD4+ T in the blood compared to the control. In addition, this is related to the IL-1 beta and IL-6. Could the author explain more on this part, especially at what stage of these findings were conducted? Because the IL-1 beta and IL-6 are considered as the early-stage cytokine according to the other parts in this review, are there T cell findings that are also early-stage inflammatory changes?

4: In lines 369-372, the author stated that TNF-a and IL-6 levels were upregulated in both postmortem and CSF of living subjects (according to the early part of this review, these two cytokines are advanced-stage proinflammatory cytokine in DLB) but not in PDD. We wonder whether this comparison happened between DLB and PDD at the same stage. Whether PDD also have early vs late-stage differences in cytokine profiles? The author later suggested the synuclein, tau, and Abeta fibrils interaction. It would be cool to know if these DLB-related TNF-a and IL-6 increase is related to new strains of synuclein fibrils by tau or A beta interaction. 

5: In lines 396-400, the author mentioned one longitudinal cohort study that assessed the blood cytokine profiles over the disease stage in DLB—the found decreasing trend in IL-1beta, IL-2, and IL-4. Do they replicate the TNFa and IL-6 increase in a later stage?

6: In lines 471-480, the author introduced small serum extracellular vesicle studies related to CNS-derived proinflammatory RNA analysis. However, we don't know what stage of the patient for this study. According to the pro-inflammatory profile, it looks like in an advanced stage of DLB patients.

7: We believe the author largely ignores the possibility of peripheral infiltrated macrophage-derived microglia in DLB patients. It has been proved that the peripheral monocyte could replenish the CNS microglia (1), and more and more single-cell studies show highly variable microglia phenotypes in animal models (2). For example, a small group of CNS immune cells called border-associated macrophages was believed to have a unique role in the initiation of CD4+ T cells response and less resolved neuroinflammation, which might be very important to the chronic stage of T cell sustain and activation in DLB patients (3). In the view of genetic factors, the PD highly related LRRK2 is also found to be expressed higher in peripheral monocyte/macrophage than CNS resident microglia, and its pathological mutation was found related to more infiltration of peripheral monocytes into the brain with alpha-synuclein pathologies (4). This is worth mentioning in the review, making readers realize the variability of microglia/macrophage phenotypes in DLB patients of different genetic backgrounds.

(1)Peripherally derived macrophages can engraft the brain independent of irradiation and maintain an identity distinct from microglia | Journal of Experimental Medicine | Rockefeller University Press (rupress.org)

(2)Frontiers | Single-Cell Transcriptomics and In Situ Morphological Analyses Reveal Microglia Heterogeneity Across the Nigrostriatal Pathway (frontiersin.org)

(3)  Border-associated macrophages mediate the neuroinflammatory response in an alpha-synuclein model of Parkinson disease | Nature Communications

(4)Pathological α-synuclein recruits LRRK2 expressing proinflammatory monocytes to the brain | Molecular Neurodegeneration | Full Text (biomedcentral.com)
